# LOCAL SGD MEETS ASYNCHRONY

## ABSTRACT

Distributed variants of stochastic gradient descent (SGD) are central to training deep neural networks on massive datasets. Several scalable versions of data-parallel SGD have been developed, leveraging asynchrony, communication-compression, and local gradient steps. Current research seeks a balance between distributed scalability–seeking to minimize the amount of synchronization needed–and generalization performance–seeking to achieve the same or better accuracy relative to the sequential baseline. However, a key issue in this regime is largely unaddressed: if "local" data-parallelism is aggressively applied to better utilize the computing resources available with workers, generalization performance of the trained model degrades.

In this paper, we present a method to improve the "local scalability" of decentralized SGD. In particular, we propose two key techniques: (a) shared-memory based asynchronous gradient updates at decentralized workers keeping the local minibatch size small, and (b) an asynchronous non-blocking in-place averaging overlapping the local updates, thus essentially utilizing all compute resources at all times without the need for large minibatches. Empirically, the additional noise introduced in the procedure proves to be a boon for better generalization. On the theoretical side, we show that this method guarantees ergodic convergence for non-convex objectives, and achieves the classic sublinear rate under standard assumptions. On the practical side, we show that it improves upon the performance of local SGD and related schemes, without compromising accuracy.

## 1 INTRODUCTION

In this paper, we consider the classic problem of minimizing an empirical risk, defined simply as

$$\min_{x \in \mathbb{R}^d} \sum_{i \in [I]} f_i(x), \tag{1}$$

where $d$ is the dimension, $x \in \mathbb{R}^d$ denotes the set of model parameters, $[I]$ is the training set, and $f_i(x) : \mathbb{R}^d \to \mathbb{R}$ is the loss on the training sample $i \in [I]$. Stochastic gradient descent (SGD) (Robbins & Monro, 1951) is an extremely popular iterative approach to solving this problem:

$$x_{k+1} = x_k - \alpha_k \nabla f_{B_k}(x_k), \tag{2}$$

where $\nabla f_{B_k}(x_k) = \frac{1}{|B_k|} \sum_{i \in B_k} \nabla f_i(x_k)$ is the sum of gradients computed over samples, typically selected uniformly and randomly as a minibatch $B_k \subseteq [I]$, and $\alpha_k$ is the learning rate at iteration $k$.

### 1.1 BACKGROUND ON DECENTRALIZED DATA-PARALLEL SGD

For better or worse, SGD and its variants currently represent the computational backbone for many large-scale optimization tasks, most notably the training of deep neural networks (DNNs). Arguably the most popular SGD variant is *minibatch SGD* (MB-SGD) (Bottou (2012)). In a distributed setting with decentralized workers $q \in [Q]$, it follows the iteration

$$x_{k+1} = x_k - \alpha_k \frac{1}{Q} \sum_{q=1}^{Q} \nabla f_{B_k^q}, \tag{3}$$

where $B_k^q \subseteq [I]$ is a local minibatch selected by worker $q \in [Q]$ at iteration $k$. This strategy is straightforward to scale in a data-parallel way, as each worker can process a subset of the samples in parallel, and the model is then updated by the average of the workers' gradient computations. For convenience, we assume the same batch size per worker. This approach has achieved tremendous popularity recently, and there has been significant interest in running training with increasingly large batch sizes aggregated over a large number of GPUs, e.g. Goyal et al. (2017).

An alternative approach is *parallel or local SGD* (L-SGD) (Zinkevich et al. (2010); Zhang et al. (2016c); Lin et al. (2020)):

$$x_{j,t+1}^q = x_{j,t}^q - \alpha_{j,t} \nabla f_{B_{j,t}^q}, \ 0 \le t < K_j; \ x_{j+1,0}^q = \frac{1}{Q} \sum_q x_{j,K_j}^q, \tag{4}$$

where $x_{j,t}^q$ denotes the local model at worker $q \in [Q]$ after $j$ synchronization rounds followed by $t$ local gradient updates and $B_{j,t}^q$ is the local minibatch sampled at the same iteration. $K_j$ denotes the number of local gradient update steps before the $j^{th}$ synchronization. Essentially, workers run SGD without any communication for several local steps, after which they globally *average* the resulting local models. This method is intuitively easy to scale, since it reduces the *frequency* of the communication. Recently, a variant called *post local SGD* (PL-SGD) (Lin et al. (2020)), was introduced to address the issue of loss in generalization performance of L-SGD, wherein the averaging frequency during the initial phase of training is high and is reduced later when optimization stabilizes.

| Method | $B_{loc}$ | Train Loss | Train Acc. | Test Loss | Test Acc | Time (Sec) | Quality/ Perf. |
|--------|-----------|------------|------------|-----------|----------|------------|----------------|
| MB-SGD | 128 | 0.016 | 99.75 | 0.234 | 92.95 | 1754 | Baseline |
| MB-SGD | 1024 | 0.023 | 99.51 | 0.293 | 91.38 | 1201 | OK |
| PL-SGD | 128 | 0.018 | 99.69 | 0.245 | 92.98 | 1603 | Good |
| PL-SGD | 1024 | 0.154 | 94.69 | 0.381 | 87.81 | 1159 | Poor |

**Table 1:** RESNET-20/CIFAR-10 training for 300 epochs on 2 GPUs. Throughout the training the local BS is kept constant across workers i.e. $|B_k^q| = B_{loc}, \forall q \in [Q], k \geq 0$ for MB-SGD and $|B_{j,t}^q| = B_{loc}, \forall q \in [Q], j, t \geq 0$ for PL-SGD. The LR is warmed up for the first 5 epochs to scale from $\underline{\alpha_0}$ to $\underline{\alpha_0} \times \frac{B_{loc} \times Q}{\underline{B_{loc}}}$, where $\underline{B_{loc}} = 128$, $Q$ is the number of workers, 2 here, and $\underline{\alpha_0} = 0.1$. In PL-SGD, we average the model after each gradient update for first 150 epochs and thereafter averaging frequency $K$ is set to 16 as in Lin et al. (2020); other HPs are identical to theirs. The listed results are average of 3 runs with different seeds.

Although very popular in practice, these two approaches suffer from the same limitation: their generalization accuracy decreases for larger *local* batch sizes, as would be appropriate to fully utilize the computation power offered by the workers. We illustrate this in Table 1: examine the results of training RESNET-20 (He et al. (2016)) over CIFAR-10 (Krizhevsky (2009)) with MB-SGD and PL-SGD over a workstation packing two Nvidia GeForce RTX 2080 Ti GPUs (a current standard). We observe that as the local batch size $B_{loc}$ grows, the throughput improves significantly, however, the optimization results degrade sharply, more glaringly with PL-SGD. Clearly, these methods can not tolerate a larger $B_{loc}$, though the GPUs can support them. This shortcoming of the existing methods in harnessing the growing data-parallelism is also identified via empirical studies (Golmant et al. (2018); Shallue et al. (2019)) existing in literature. To our knowledge no effective remedy (yet) exists to address this challenge.

Notice that, here our core target is maximally harnessing the *local* data-parallelism and therefore the larger *local* batch size, as against the existing trend in the literature wherein large number of GPUs are deployed to have a large aggregated *global* batch size with a relatively small $B_{loc}$. For example, refer to the performance of MB-SGD and PL-SGD as listed in Table 1 of Lin et al. (2020). Notice that with 16 GPUs, each with $B_{loc} = 128$, thus totaling the minibatch size as 2048, identical to the one with 2 GPUs each with $B_{loc} = 1024$ as above, with exactly the same LR scaling and warmup strategy, both MB-SGD and PL-SGD do not face generalization degradation. However, unfortunately, such an implementation setting would incur excessive wastage of available data-parallel compute resources on each of the GPUs. Indeed, the existing specific techniques such as LARS (You et al. (2017)) to address the issue of poor generalization for global large batch training are insufficient for the larger local minibatch size; we empirically describe it in Section 3 (Table 11).

## 1.2 LOCALLY-ASYNCHRONOUS PARALLEL SGD

Now, consider an implementation scheme as the following:

1. In a decentralized setting of L-SGD, i.e. wherein each worker $q \in [Q]$ has a local model $x^q$ undergoing local SGD updates as described earlier, multiple *local concurrent processes* $u \in U^q$ share the model $x^q$. Processes $u \in U^q$ perform asynchronous concurrent gradient updates locally.
2. The workers average their models whenever any one of them would have had at least $K_j$ local shared updates, where $K_j$ is as that in Equation 4. The averaging is performed asynchronously and in a non-blocking way by the *(averaging-) processes* $a^q$ on behalf of each worker $q \in [Q]$.

Essentially, the decentralized workers run shared-memory-based asynchronous SGD locally and periodically synchronize in a totally non-blocking fashion.

More formally, consider Algorithm 1. The model $x^q$ on a GPU $q \in [Q]$ is shared by the processes $p \in P^q = \{\{a^q\} \cup U^q\}$ locally. The processes $p \in P^q$ also maintain a *shared counter* $\mathcal{S}^q$, initialized to 0. The operation `read-and-inc` implements an atomic (with lock) read and increment of $\mathcal{S}^q$, whereas, `read` provides an atomic read. $\mathcal{S}^q$ essentially enables ordering the shared gradient updates. In turn, this order streamlines the synchronization among workers, thereby determines the averaging rounds $j$. The *(updater) processes* $u \in U^q$ asynchronously and lock-freely update $x^q$ with

gradients computed over a non-blocking, potentially inconsistent, snapshot $v^{a,q}$ of $x^q$, essentially going Hogwild! (Recht et al. (2011)), see Algorithm 1a.

---

1 Initialize $s = 0$;
2 **while** $s \leq T$ **do**
3    $v^{u,q}[i] := x^q[i], \forall\, 1 \leq i \leq d$;
4    $s := \texttt{read-and-inc}(\mathcal{S})$;
5    Compute $\nabla f_{B_s^q}(v^{u,q})$;
6    $x^q[i] \mathrel{-}= \alpha_s \nabla f_{B_s^q}(v^{u,q})[i]$,
     $\forall\, 1 \leq i \leq d$;

(a) Local asynchronous gradient update by process $u \in U^q$.

1 Initialize $s_{cur} = s_{pre} = |U^q|, j = 0$;
2 **while** $s_{cur} \leq T$ **do**
3    $s_{cur} := \texttt{read}(\mathcal{S})$; Compute $j$ corresponding to $s_{cur}$;
4    **if** $s_{cur} - s_{pre} \geq K_j$ **then**
5      $v_j^{a,q}[i] := x^q[i], \forall\, 1 \leq i \leq d$;
6      Synchronize across $a^r, r \in [Q] \setminus \{q\}$ to compute
       $\overline{v_j} := \frac{1}{Q} \sum_{q \in [Q]} v_j^{a,q}$;
7      Compute $\Delta v_j^q = \overline{v_j} - v_j^{a,q}$; $s_{pre} := s_{cur}$;
8      $x^q[i] \mathrel{+}= \Delta v_j^q[i], \forall\, 1 \leq i \leq d$; $j = j + 1$;

(b) Asynchronous non-blocking in-place averaging.

Algorithm 1: Locally-asynchronous Parallel SGD (Lap-SGD)

The process $a^q$, which performs *averaging* for the worker $q \in [Q]$, concurrently keeps on atomically reading $\mathcal{S}^q$, see Algorithm 1b. As soon as it notices an increment $K_j$ in $\mathcal{S}^q$, i.e. $x^q$ got concurrently updated with $K_j$ number of gradients , it takes a non-blocking snapshot $v_j^{a,q}$ of $x^q$ and synchronizes with $a^r$ of peers $r \in [Q]/q$ to compute the average $\overline{v_j}$ of the snapshots. Thereafter, $a^q$ adds the difference of the average with the snapshot $v_j^{a,q}$ to the model $x^q$ without blocking the concurrent asynchronous local gradient updates. We call this method *locally-asynchronous parallel SGD* (Lap-SGD). This method closely resembles Hogwild++ (Zhang et al. (2016a)), which targets the heterogeneous NUMA based multi-core machines, though there are key differences which we describe in Section 4. Results of the same training task as before by Lap-SGD is given in Table 2. The distinction of this implementation is that it harnesses the compute power of the GPUs not by increasing the size of $B_{loc}$ but by concurrently computing many minibatch gradients. Evidently, Lap-SGD provides speed-up without losing the quality of optimization in comparison to the baseline.

Recently, Kungurtsev et al. (2019) presented a shared-memory based method wherein they showed that partitioned gradient updates for some iterations during the course of training over a shared model can reasonably save on total computation cost by means of restricted backpropagation without necessarily losing on optimization quality. Their method is limited to a centralized shared-memory setting. Moreover, aiming to establish

| $|U^q|$ | $B_{loc}$ | Train Loss | Train Acc. | Test Loss | Test Acc | Time (Sec) | Quality/ Perf. |
|---|---|---|---|---|---|---|---|
| 4 | 128 | 0.012 | 99.83 | 0.270 | 92.92 | 1304 | Excellent |
| 6 | 128 | 0.023 | 99.51 | 0.266 | 92.91 | 1281 | Excellent |
| 4 | 256 | 0.010 | 99.90 | 0.280 | 92.93 | 1219 | Excellent |

Table 2: ResNet-20/Cifar-10 training for 300 epochs on 2 GPUs by Lap-SGD. The LR is warmed up similar to PL-SGD.We follow the same averaging schedule and other HPs as those of PL-SGD. The listed results are average of 3 runs.

convergence under non-smoothness assumption, they ensure write consistency under a model-wide lock. Having designed our asynchronous parallel SGD, it inspires us to adapt the partitioned gradient update strategy to our lock-free decentralized setting.

More specifically, building on Lap-SGD, we consider locally partitioned gradient computation along with asynchronous lock-free updates. Essentially, we partition the model $x^q$ to $\{x_{i(u)}^q\}$ for $u \in U^q$, $i(u) \cap i(w) = \emptyset, \forall u, w \in U^q$ (i.e., non-overlapping block components of the vector $x$). With that, a partitioned gradient computation

| $|U^q|$ | $B_{loc}$ | Train Loss | Train Acc. | Test Loss | Test Acc | Time (Sec) | Quality/ Perf. |
|---|---|---|---|---|---|---|---|
| 4 | 128 | 0.019 | 99.62 | 0.270 | 92.98 | 1153 | Excellent |
| 6 | 128 | 0.021 | 99.58 | 0.262 | 92.84 | 1085 | Excellent |
| 4 | 256 | 0.019 | 99.65 | 0.267 | 92.95 | 1047 | Excellent |

Table 3: Lpp-SGD performance. Other details are identical to those in Table 2.

will amount to computing $\overline{\nabla}_{i(u)} f_{B_s^q}(v^{q,u})$, the minibatch gradient with respect to the partition $x_{i(u)}^q$ at line 5 in Figure 1a. Accordingly, the update step at line 6 in Algorithm 1a transforms to $x^q[i] \mathrel{-}= \alpha_s \nabla f_{B_s^q}(v^{q,u})[i], \forall\, i \in i(u)$. It is to be noted that we do not use write lock for iterations at any stage. Having devised a partitioned update scheme, we propose *locally-partitioned-asynchronous parallel SGD* (Lpp-SGD) as described below.

1. Processes $u \in U^q$ maintain a process-local variable $\texttt{last\_iter}$ which can take two values PARTITIONED and FULL. Each $u \in U^q$ initializes $\texttt{last\_iter}$ as FULL.
2. **While** $s \leq T_{st}$, each process $u \in U^q$ performs Lap-SGD updates as lines 3 to 6 of Algorithm 1a.
3. **If** $T_{st} < s \leq T$, each process $u \in U^q$ performs

    (a) a partitioned gradient computation and update: $x^{q,u}[i] \mathrel{-\!\!=} \alpha_s \nabla f_{B_s^q}(v^{u,q})[i], \forall i \in i(u)$ **if** `last_iter` = FULL, and sets `last_iter` = PARTITIONED.

    (b) an LAP-SGD update **if** `last_iter` = PARTITIONED, and sets `last_iter` = FULL.

Essentially, after some initial *stabilizing* epochs each process $u \in U^q$ alternates between a full and a partitioned lock-free asynchronous gradient updates to the model $x^q$. Our experiments showed that $T_{st} = \frac{T}{10}$ was almost always sufficient to obtain a competitive optimization result. The results of a sample implementation of LPP-SGD are available in Table 3. It is clear that LPP-SGD handsomely speeds up the computation and provides equally competitive optimization results.

## 2 CONVERGENCE THEORY

At a naive first glance, studying the convergence properties of *locally asynchronous SGD* would be an incremental to existing analyses for local SGD, e.g. Stich (2018); Zhou & Cong (2017), in particular the local stochastic gradient evaluations at delayed lock-free Hogwild!-like parameter vectors. However, there is one significant difference that presents a theoretical challenge: sometimes the vectors used for gradient computation or components thereof, have been read from the local shared memory before the last averaging across GPUs had taken place. Especially in the nonconvex case, *a priori* it is impossible to place a reasonable bound on the accuracy of these gradient evaluations relative to what they "should be" in order to achieve descent.

---

1   Initialize $\bar{x}_0 = x_0^q$ for all $q$;
2   **for** $j = 1, ..., J$ **do**
3     **for** *all* $q$ **do**
4       Set $x_0^{q,j} = \bar{x}_j$;
5       **for** $t = 1, ..., K_j$ **do**
6         Let $x_t^{q,j} = x_{t-1}^{q,j} -$
         $\alpha_{j,t,q}\tilde{\nabla}_{(i(j,t,q)}f(v_t^{q,j})$;
7     Let $\bar{x}_{j+1} = \frac{1}{Q}\sum\limits_{q=1}^{Q} x_{K_j}^{q,j}$;

---

Algorithm 2: Iterations of the view $\bar{x}_j$.

In order to present a convergence rate result, we need to define an anchor point on which we can consider convergence to some approximate stationary point in expectation. This is not trivial as both local iterations and averaging is performed asynchronously across different GPUs at distinct moments in time, with the time at which each iteration occurs potentially varying with system-dependent conditions, while the averaged iterates are what is important to consider for convergence.

We seek to define the *major iteration* $\bar{x}_j$ as consistent with the analysis presented for the convergence of local SGD in the nonconvex setting in Zhou & Cong (2017). In this case, with asynchrony, $\bar{x}_j$ is a theoretical construct, i.e., it may not exist at any particular GPU's memory at any moment in time. Let $s_j^q := s_{cur} - |U^q|$ be the current state of the shared counter before the $j^{th}$ synchronization round at the GPU $q$, then $\bar{x}_j$ is defined as $x_{s_j^q}^q + \Delta v_j$ where $x_{s_j^q}^q$ is the state of the model for GPU $q$ when $v_j^{a,q}$ was saved and made available for averaging for "major iteration" $j$. Thus although de facto averaging could have taken place after a set of local updates in time, these local updates correspond to updates after iteration $j$ conceptually. This makes $\bar{x}_j$ properly correspond to the equivalent iterate in Zhou & Cong (2017).

With that, we consider a sequence of *major iteration views* $\{\bar{x}_j\}$ with associated inter-averaging local iteration quantities $K_j$ and *local views* $\{v_t^{q,j}\}$ at which an unbiased estimate of the (possibly partitioned) gradient is computed, with $0 \le t < K_j$ as well as the local model in memory $\{x_t^{q,j}\}$. The partition, which could be the entire vector in general, is denoted by $i(j,t,q)$. As each GPU has its own corresponding annealed stepsize, we denote it in general as $\alpha_{j,t,q}$ as well.

We state the formal mathematical algorithm as Algorithm 2. Note that this is the same procedure as the practical "what the processor actually does" Algorithm 1, however, with the redefined terms in order to obtain a precise mathematical sequence well-defined for analysis.

We make the following standard assumptions on unbiased bounded variance for the stochastic gradient, a bounded second moment, gradient Lipschitz continuity, and lower boundedness of $f$.

**Assumption 2.1.** *1. It holds that $\tilde{\nabla}_i f(v_t^{q,j})$ satisfies, independent of $i$,* $\mathbb{E}\left[\tilde{\nabla}_i f(v_t^{q,j})\right] =$

$\nabla_i f(v_t^{q,j}); \mathbb{E}\left[\left\|\tilde{\nabla}_i f(v_t^{q,j}) - \nabla_i f(v_t^{q,j})\right\|^2\right] \le \sigma^2; \mathbb{E}\left[\left\|\tilde{\nabla}_i f(v_t^{q,j})\right\|^2\right] \le G.$

*2. $f$ is Lipschitz continuously differentiable with constant $L$ and is bounded below by $f_m$.*

We must also define some assumptions on the probabilistic model governing the asynchronous computation. As these are fairly technical we defer them to the appendix.

**Theorem 2.1.** *Given assumption 2.1, it holds that,*

$$
\begin{aligned}
\frac{1}{Q}\sum_{j=1}^{J}\sum_{q=1}^{Q}\sum_{t=0}^{K_j-1}[\alpha_{j,t,q}C_1 - \alpha_{j,t,q}^2 C_2]\mathbb{E}\left\|\nabla_{i(j,t,q)}f(v_t^{q,j})\right\|^2 & \\
+\frac{1}{Q}\sum_{j=1}^{J}\sum_{q=1}^{Q}\sum_{t=0}^{K_j-1}[\alpha_{j,t,q}C_3 - \alpha_{j,t,q}^2 C_4]\mathbb{E}\left\|\nabla_{i(j,t,q)}f(\bar{x}_j)\right\|^2 & \\
\leq f(\bar{x}_0) - f_m &
\end{aligned}
\tag{5}
$$

*where $C_1$, $C_2$, $C_3$, and $C_4$ depend on L, B and probabilistic quantities defining the asynchronous computation (see Appendix. Thus there exists a set of such constants such that if $\alpha_{j,t,q} = \Theta\left(\frac{1}{\sqrt{J}}\right)$ then Algorithm 2 ergodically converges with the standard $O(1/\sqrt{J})$ rate for nonconvex objectives.*

*Proof Summary:* The proof follows the structure of the ergodic convergence proof of K-step local SGD given in Zhou & Cong (2017), wherein at each round of averaging there are $QK_j$ total updates to the model vector associated with the $Q$ GPUs and $K_j$ minor iterations.

Insofar as these updates are close (stochastically as an unbiased estimate, and based on the local models not having changed too much) to the globally synchronized model vector at the last averaging step, there is an expected amount of descent achieved due to the sum of these steps. This is balanced with the amount of possible error in this estimate based on how far the model vector had moved.

In cases wherein $v_{t,i}^{q,j} = x_{s,i}^{q,j}$ for $s < 0$ (i.e., the stochastic gradients are taken, due to local asynchrony, at model vectors with components which existed in memory before the last averaging step), we simply bound the worst-case increase in the objective.

To balance these two cases, the analysis takes an approach, inspired partially by the analysis given in Cartis & Scheinberg (2018) of separating these as "good" and "bad" iterates, with "good" iterates corresponding to views read after the last model was stored for averaging, with some associated guaranteed descent in expectation, and "bad" iterates those read beforehand.

By considering the stochastic process governing the amount of asynchrony as being governed by probabilistic laws, we can characterize the probability of a "good" and "bad" iterate and ultimately seek to balance the total expected descent from one, and worst possible ascent in the other, as a function of these probabilities.

**Remark 2.1.** [**Speedup due to concurrent updates**] Consider the case of classical vanilla local SGD, in which there is complete symmetry in the number of local gradient computations between averaging steps and block sizes across the processors. In this case, for every major iteration there are $Q$ gradient norms on the left hand side, and at the same time it is divided by $Q$. Thus local SGD as a parallel method does not exhibit classical speedup, rather it can be considered as an approach of using parallelism to have a more robust and stable method of performing gradient updates with multiple batches computed in parallel. However, due to the idle time that exists between the slowest and fastest processors, it will exhibit *negative* speedup, relative to the fastest processor. With the approach given in this paper, this negative speedup is corrected for in that this potential idleness is filled with additional stochastic gradient computations by the fastest process. Alternatively, one can also consider this as exhibiting positive speedup relative to the slowest process, whereas standard local SGD has zero speedup relative to the slowest process. Above and beyond this, we can consider that as more processes introduces additional latency and delay, which has a mixed effect: on the one hand, we expect that gradient norms at delayed iterates to be larger as the process is converging, thus by having more delayed gradients on the left hand side, convergence is faster, and on the other hand, such error in the degree to which the gradient decreases the objective, would specifically increase the constants $C_2$ and $C_4$.

## 3 IMPLEMENTATION DETAILS AND NUMERICAL RESULTS

### 3.1 EXPERIMENTAL SET-UP

**Decentralized training.** We evaluate the proposed methods LAP-SGD and LPP-SGD comparing them against existing MB-SGD and PL-SGD schemes, using CNN models RESNET-20 (He et al. (2016)), SQUEEZENET (Iandola et al. (2017)), and WIDERESNET-16x8 (Zagoruyko & Komodakis (2016)) for the 10-/100-class image classification tasks on datasets CIFAR-10/CIFAR-100 (Krizhevsky (2009)). We also train RESNET-18 for a 1000-class classification problem on IMAGENET (Russakovsky et al. (2015)) dataset. We keep the sample processing budget identical across the methods. We use the typical approach of partitioning the sample indices among the workers that can access the entire training set; the partition indices are reshuffled every epoch following a seeded

random permutation based on epoch-order. To this effect we use a shared-counter among concurrent process $u \in U^q$ in asynchronous methods. Thus, our data sampling is i.i.d.

**Platform specification.** Our experiments are based on a set of Nvidia GeForce RTX 2080 Ti GPUs (Nvidia (2020)) (referred to as GPUs henceforth) with 11 GB on-device memory. We use the following specific settings: (a) **S1:** a workstation with two GPUs and an Intel(R) Xeon(R) E5-1650 v4 CPU running @ 3.60 GHz with 12 logical cores, (b) **S2:** a workstation with four GPUs and two Intel(R) Xeon(R) E5-2640 v4 CPUs running @ 2.40 GHz totaling 40 logical cores, and (c) **S3:** two **S2** workstations connected with a 100 GB/s infiniband link. The key enabler of our implementation methodology are multiple independent client connection between a CPU and a GPU. Starting from early 2018 with release of Volta architecture, Nvidia's technology Multi-process Service (MPS) efficiently support this. For more technical specifications please refer to their doc-pages MPS (2020).

**Implementation framework.** We used open-source Pytorch 1.5 (Paszke et al. (2017)) library for our implementations. For cross-GPU/cross-machine communication we use NCCL (NCCL (2020)) primitives provided by Pytorch. MB-SGD is based on `DistributedDataParallel` Pytorch module. PL-SGD implementation is derived from author's code (LocalSGD (2020)) and adapted to our setting. Having generated the computation graph of the loss function of a CNN, the `autograd` package of Pytorch allows a user to specify the leaf tensors with respect to which gradients are needed. We used this functionality in implementing partitioned gradient computation in LPP-SGD.

**Locally-asynchronous Implementation.** One key requirement of the proposed methods is to support a non-blocking synchronization among GPUs. This is both a challenge and an opportunity. To specify, we use a process on each GPU, working as a parent, to initialize the CNN model and share it among spawned child-processes. The child-processes work as $u \in U^q$, $\forall q \in [Q]$ to compute the gradients and update the model. Concurrently, the parent-process, instead of remaining idle as it happens commonly with such concurrency models, acts as the averaging process $a^q \in P^q$, $\forall q \in [Q]$, thereby productively utilizing the entire address space occupied over the GPUs. The parent- and child-processes share the iteration and epoch counters. Notice that, here we are using the process-level concurrency which is out of the purview of the thread-level global interpreter lock (GIL) (Python (2020)) of python multi-threading framework.

**Hyperparameters (HPs).** Each of the methods use identical momentum (Sutskever et al. (2013)) and weight-decay (Krogh & Hertz (1991)) for a given CNN/dataset case; we rely on their previously used values (Lin et al. (2020)). The learning rate (LR) schedule for MB-SGD and PL-SGD are identical to Lin et al. (2020). For the proposed methods we used cosine annealing schedule without any intermediate restart (Loshchilov & Hutter (2017)). Following the well-accepted practice, we *warm up* the LR for the first 5 epochs starting from the baseline value used over a single worker training. In some cases, a grid search (Pontes et al. (2016)) suggested that for LPP-SGD warming up the LR up to $1.25\times$ of the *warmed-up* LR of LAP-SGD for the given case, improves the results.

### 3.2 EXPERIMENTAL RESULTS

In the following discussion we use these abbreviated notations: $\mathbf{U}$ : $|U^q|$, $\mathbf{B}$ : $B_{loc}$, Tr.L.: Training Loss, Tr.A.: Training Accuracy, Te.L.: Test Loss, Te.A.: Test Accuracy, and $T$: Time in Seconds. The asynchronous methods have inherent randomization due to process scheduling by the operation system. Therefore, each micro-benchmark presents the mean of 3 runs unless otherwise mentioned.

| U | B | Tr.A. | Te.A. | T |
|---|---|---|---|---|
| 4 | 128 | 99.62 | 92.81 | 1304 |
| 6 | 128 | 99.85 | 92.92 | 1266 |
| 8 | 128 | 99.62 | 92.64 | 1259 |
| 4 | 256 | 99.72 | 92.83 | 1219 |
| 6 | 256 | 99.90 | 92.24 | 1166 |

Table 4: LAP-SGD performance.

**Concurrency on GPUs.** We allocate the processes on a GPU up to the availability of the on-device memory. However, once the data-parallel compute resources get saturated, allocating more processes degrades the performance. For example, see Table 4 which lists the average performance of 5 runs for different combinations of $\mathbf{U}$ and $\mathbf{B}$ for training RESNET-20/CIFAR-10 by LAP-SGD on the setting **S1**.

**Asynchronous Averaging Frequency.** Following PL-SGD, as a general rule, for the first half of the training, call it up until $\mathbf{P}$ epochs, we set the averaging frequency $\mathbf{K} = 1$. However, notice that, unlike PL-SGD, setting a $\mathbf{K} < Q$ in LAP-SGD and LPP-SGD may not necessarily increase the averaging rounds $j$ in aggregation. Intuitively, in a locally-asynchronous setting, along with the non-blocking (barrier-free) synchronization among GPUs, the increment events on the shared-counter $\mathcal{S}$ would be "grouped" on the real-time scale if the processes $u \in U^q$ do not face delays in scheduling, which we control by allocating an optimal number of processes to maximally utilize the compute

resources. For instance, Table 5 lists the results of 5 random runs of RESNET-20/CIFAR-10 training with $\mathbf{B} = 128$ and $\mathbf{U} = 6$ with different combinations of $\mathbf{K}$ and $\mathbf{P}$ over $\mathbf{S1}$. This micro-benchmark indicates that the overall latency and the final optimization result of our method may remain robust under small changes in $\mathbf{K}$, which it may not be the case with PL-SGD.

**Scalability.** Table 6 presents the results of WIDERESNET-16x8/CIFAR-10 training in the three system settings that we consider. We observe that in each setting the relative speed-up of different methods are approximately consistent. In particular, we note the following: (a) reduced communication cost helps PL-SGD marginally outperform MB-SGD, (b) interestingly, increasing $B_{loc}$ from 128 to 512 does not improve the latency by more than $\sim$4%; this non-linear

| K | P | Tr.A. | Te.A. | $T$ |
|---|---|-------|-------|-----|
| 1 | 150 | 99.76 | 93.01 | 1266 |
| 4 | 150 | 99.85 | 92.78 | 1263 |
| 8 | 150 | 99.62 | 93.08 | 1262 |
| 8 | 0 | 99.78 | 92.83 | 1263 |
| 16 | 150 | 99.90 | 93.16 | 1261 |

Table 5: LAP-SGD performance.

scalability of data-parallelism was also observed by Lin et al. (2020) , (c) each of the methods scale by more than 2x as the implementation is moved to $\mathbf{S2}$, which has 40 CPU cores, from $\mathbf{S1}$ which has 12 CPU cores, furthermore, this scalability is approximately 2x with respect to performance on $\mathbf{S3}$ in comparison to $\mathbf{S2}$, this shows that for each of the methods we utilize available compute resources maximally, (d) in each setting LAP-SGD achieves $\sim$30% better throughput compared to MB-SGD standard batch, (e) in each setting LPP-SGD outperforms LAP-SGD by $\sim$12% making it the fastest method, (f) the training and test accuracy of local large minibatch is poor, and (g) the methods LAP-SGD and LPP-SGD consistently improve on the baseline generalization accuracy.

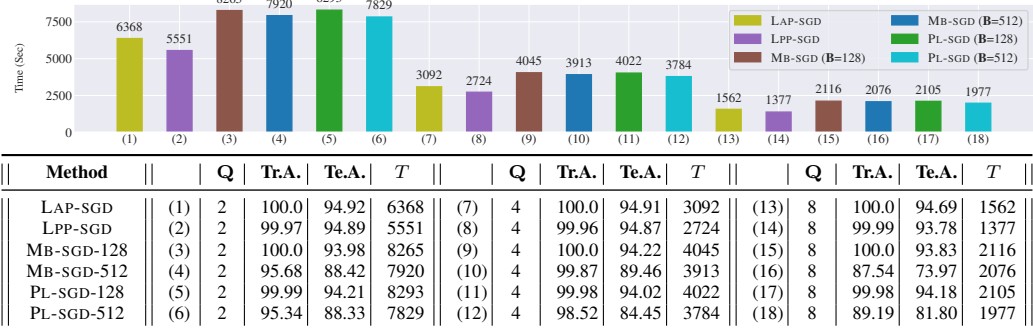

| Method | | | Q | Tr.A. | Te.A. | $T$ | | Q | Tr.A. | Te.A. | $T$ | | Q | Tr.A. | Te.A. | $T$ | |
|--------|---|---|---|-------|-------|-----|---|---|-------|-------|-----|---|---|-------|-------|-----|---|
| LAP-SGD | (1) | | 2 | 100.0 | 94.92 | 6368 | (7) | 4 | 100.0 | 94.91 | 3092 | (13) | 8 | 100.0 | 94.69 | 1562 | |
| LPP-SGD | (2) | | 2 | 99.97 | 94.89 | 5551 | (8) | 4 | 99.96 | 94.87 | 2724 | (14) | 8 | 99.99 | 93.78 | 1377 | |
| MB-SGD-128 | (3) | | 2 | 100.0 | 93.98 | 8265 | (9) | 4 | 100.0 | 94.22 | 4045 | (15) | 8 | 100.0 | 93.83 | 2116 | |
| MB-SGD-512 | (4) | | 2 | 95.68 | 88.42 | 7920 | (10) | 4 | 99.87 | 89.46 | 3913 | (16) | 8 | 87.54 | 73.97 | 2076 | |
| PL-SGD-128 | (5) | | 2 | 99.99 | 94.21 | 8293 | (11) | 4 | 99.98 | 94.02 | 4022 | (17) | 8 | 99.98 | 94.18 | 2105 | |
| PL-SGD-512 | (6) | | 2 | 95.34 | 88.33 | 7829 | (12) | 4 | 98.52 | 84.45 | 3784 | (18) | 8 | 89.19 | 81.80 | 1977 | |

Table 6: WIDERESNET-16x8/CIFAR-10 training in the settings $\mathbf{S1}$: $\mathbf{Q} = 2$, $\mathbf{S2}$: $\mathbf{Q} = 4$, and $\mathbf{S3}$: $\mathbf{Q} = 8$. MB-SGD-128 indicates MB-SGD method with $B_{loc} = 128$ and similarly for others. For LAP-SGD, we spawned 4 concurrent processes on each GPU: $\mathbf{U} = 4$, whereas for LPP-SGD $\mathbf{U} = 3$ concurrent processes were spawned.

| Method | B | U | Tr.L. | Te.L. | Tr.A. | Te.A. | $T$ | Method | B | U | Tr.L. | Te.L. | Tr.A. | Te.A. | $T$ | |
|--------|---|---|-------|-------|-------|-------|-----|--------|---|---|-------|-------|-------|-------|-----|---|
| LAP-SGD | 128 | 6 | 0.295 | 1.191 | **92.23** | **69.43** | 1285 | MB-SGD | 1024 | - | 0.464 | 1.212 | 86.46 | 67.04 | 1244 | |
| LAP-SGD | 128 | 4 | 0.299 | 1.198 | **92.15** | **69.78** | 1325 | MB-SGD | 128 | - | 0.360 | 1.111 | 89.93 | 69.73 | 1754 | |
| LPP-SGD | 128 | 6 | 0.404 | 1.173 | 88.69 | 69.47 | 1085 | PL-SGD | 1024 | - | 0.373 | 1.152 | 89.48 | 67.97 | 1198 | |
| LPP-SGD | 128 | 4 | 0.378 | 1.161 | 89.50 | 69.75 | 1154 | PL-SGD | 128 | - | 0.379 | 1.099 | 89.29 | 69.67 | 1613 | |

Table 7: Performance of RESNET-20 on CIFAR-100 over the setting $\mathbf{S1}$.

| Method | B | U | Tr.L. | Te.L. | Tr.A. | Te.A. | $T$ | Method | B | U | Tr.L. | Te.L. | Tr.A. | Te.A. | $T$ | |
|--------|---|---|-------|-------|-------|-------|-----|--------|---|---|-------|-------|-------|-------|-----|---|
| LAP-SGD | 64 | 4 | 0.007 | 1.341 | **99.98** | **75.97** | 1564 | MB-SGD | 512 | - | 0.034 | 1.918 | 99.64 | 63.74 | 2076 | |
| LPP-SGD | 64 | 3 | 0.008 | 1.554 | **99.97** | **75.42** | 1376 | PL-SGD | 128 | - | 0.020 | 1.354 | 99.81 | 72.38 | 2108 | |
| MB-SGD | 128 | - | 0.007 | 1.348 | 99.97 | 73.29 | 2115 | PL-SGD | 512 | - | 0.393 | 1.758 | 88.91 | 60.47 | 1976 | |

Table 8: Performance of WIDERESNET-16x8 on CIFAR-100 over the setting $\mathbf{S3}$.

| Method | B | U | Tr.L. | Te.L. | Tr.A. | Te.A. | $T$ | Method | B | U | Tr.L. | Te.L. | Tr.A. | Te.A. | $T$ | |
|--------|---|---|-------|-------|-------|-------|-----|--------|---|---|-------|-------|-------|-------|-----|---|
| LAP-SGD | 128 | 4 | 0.004 | 0.385 | **99.95** | **92.48** | 1251 | MB-SGD | 128 | - | 0.005 | 0.294 | 99.93 | 92.41 | 2243 | |
| LPP-SGD | 256 | 3 | 0.007 | 0.355 | 99.89 | **92.55** | 1027 | PL-SGD | 1024 | - | 0.005 | 0.329 | 99.93 | 91.99 | 1202 | |
| MB-SGD | 1024 | - | 0.003 | 0.325 | 99.97 | 91.50 | 1580 | PL-SGD | 128 | - | 0.006 | 0.283 | 99.91 | 92.51 | 1843 | |

Table 9: Performance of SQUEEZENET on CIFAR-10 over the setting $\mathbf{S1}$.

**Other CIFAR-10/CIFAR-100 Results.** Performance of the proposed methods in comparison to the baselines for other training tasks on CIFAR-10/CIFAR-100 datasets are available in Tables 7, 8, and 9. In each case we use $\mathbf{K} = 16$ in LAP-SGD, LPP-SGD, and PL-SGD after 50% of total sample processing budget. As an overview, the relative latency of the methods are as seen in Table 6, whereas in each case LAP-SGD and LPP-SGD recovers or improves the baseline training results.

**Imagenet Training Results.** Having comprehensively evaluated the proposed methods on CIFAR-10/CIFAR-100, here we present their performance on 1000-classes Imagenet dataset.

| | | Method | B | U | Tr.L. | Tr.A. | Te.L. | Te.A. | $T$ | |
|---|---|---|---|---|---|---|---|---|---|---|
| (1) | | Lap-sgd | 32 | 4 | 1.319 | 68.97 | 1.215 | **70.58** | **26237** | |
| (2) | | Lpp-sgd | 32 | 4 | 1.397 | 68.31 | 1.257 | 69.91 | 24066 | |
| (3) | | Mb-sgd | 128 | - | 1.237 | 70.63 | 1.220 | 70.07 | 26772 | |
| (4) | | Mb-sgd | 32 | - | 1.339 | 68.53 | 1.231 | 70.13 | 30706 | |
| (5) | | Pl-sgd | 128 | - | 1.256 | 70.22 | 1.223 | 70.02 | 25850 | |
| (6) | | Pl-sgd | 32 | - | 1.350 | 68.32 | 1.236 | 70.02 | 31468 | |

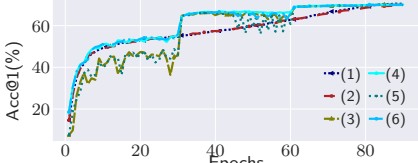

Table 10: RESNET-18/IMAGENET Training results.   Figure 2: Top-1 Test Accurcy.

Notice that for this training task, with 8 commodity GPUs at our disposal we are very much in the small minibatch setting. Plethora of existing work in the literature efficiently train a RESNET on IM-AGENET with BS up to multiple thousands. Other system-dependent constraint that our considered setting faces is that there is hardly any leftover compute resources for us to exploit in the local setting of a worker. Yet, we see for RESNET-18, see Table 10, that LAP-SGD improves on generalization accuracy of the baseline with speed-up.

**LR Tuning strategy.** It is pertinent to mention that the existing techniques, such as LARS (You et al. (2017)), which provide an adaptive LR tuning strategy for large minibatch settings over a large number of GPUs, wherein each worker locally processes a small minibatch, are insufficient in the case $B_{loc}$ is increased. For example, see Table 11 which lists the average performance of 5 runs on the setting **S1** for $Q = 2$ for training RESNET-20/CIFAR-10 using the compared methods combined with LARS with $\eta = 0.001$ (You et al. (2017)). We scale the LR proportionately: $\underline{\alpha_0} \times \frac{\mathbf{B} \times Q}{\underline{B_{loc}}}$, where $\underline{B_{loc}} = 128$, $Q$ is the number of workers, 2 here, and $\underline{\alpha_0} = 0.1$. Evidently, LARS did not help MB-SGD and PL-SGD in checking the poor generalization due to larger **B**.

| Method | U | B | Tr.A. | Te.A. | $T$ |
|---|---|---|---|---|---|
| Mb-sgd | 1 | 1024 | 95.84 | 85.67 | 1324 |
| Pl-sgd | 1 | 1024 | 93.67 | 84.69 | 1304 |
| Lap-sgd | 4 | 256 | 99.87 | 92.90 | 1232 |
| Lpp-sgd | 4 | 256 | 99.59 | 92.83 | 1071 |

Table 11: LARS performance.

### 3.3 ON THE GENERALIZATION OF THE PROPOSED METHODS

Let us comment on a theoretical basis of the remarkable performance in terms of generalization error of our methods. In Lin et al. (2020), the model of SGD algorithms as a form of a Euler-Maruyama discretization of a Stochastic Differential Equation (SDE) presents the perspective that the batch-size can correspond to the inverse of the injected noise. Whereas distributed SGD combined stochastic gradients as such effectively results in an SGD step with a larger batch-size, local SGD, by averaging models rather than gradients, maintains the noise associated with the local small-batch gradients. Given the well-established benefits of greater noise in improving generalization accuracy (e.g. Smith & Le (2018) and others), this presents a heuristic argument as to why local SGD tends to generalize better than distributed SGD. In the Appendix we present an additional argument for why local SGD generalizes well.

However, we see that our particular variation with asynchronous local updates and asynchronous averaging seems to provide additional generalization accuracy above and beyond local SGD. We provide the following explanation as to why this could be the case, again from the perspective of noise as it would appear in a SDE. Let us recall three facts,

1. The noise appearing as a discretization of the Brownian motion term in the diffusion SDE, and, correspondingly the injected noise studied as a driver for increased generalization in previous works on neural network training is i.i.d.,
2. Clearly, the covariances of a mini-batch gradients as statistical estimates of the gradient at $x$ and $x'$ are going to be more similar when $x$ and $x'$ are closer together,
3. (see Section 2) A challenging property from the perspective of convergence analysis with locally asynchronous updates is gradients taken at models taken at snapshots before a previous all-to-all averaging step, and thus far from the current model in memory.

Thus, effectively, the presence of these "highly asynchronous" stochastic gradients, while being potentially problematic from the convergence perspective, effectively brings the analogy of greater injected noise for local SGD over distributed data-parallel closer to accuracy by inducing greater probabilistic independence, i.e., the injected noise, for these updates, is far close to the i.i.d. noise that appears in a discretized SDE.

## 4 RELATED WORK

In the previous sections we cited the existing literature wherever applicable, in this section we present a brief overview of closely related works and highlight our novelty. In the shared-memory

setting, HOGWILD! (Recht et al. (2011)) is now the classic approach to implement SGD. However, it remains applicable to a centralized setting of a single worker and therefore is not known to have been practically utilized for large scale DNN training. Its success led to designs of variants which targeted specific system aspects of a multi-core machine. For example, Buckwild! (Sa et al. (2015)) proposed using restricted precision training on a CPU. Another variant, called HOG-WILD!++ (Zhang et al. (2016b)), harnesses the non-uniform-memory-access (NUMA) architecture based multi-core computers. In this method, threads *pinned* to individual CPUs on a multi-socket mainboard with access to a common main memory, form clusters.

In principle, the proposed LAP-SGD can be seen as deriving from HOGWILD!++. However, there are important differences: (a) at the structural level, the averaging in HOGWILD!++ is binary on a ring graph of thread-clusters, further, it is a token based procedure where in each round only two neighbours synchronize, whereas in LAP-SGD it is all-to-all, (b) in HOGWILD!++ each cluster maintains two copies of the model: a locally updating copy and a buffer copy to store the last synchronized view of the model, whereby each cluster essentially passes the "update" in the local model since the last synchronization to its neighbour, however, this approach has a drawback as identified by the authors: the update that is passed on a ring of workers eventually "comes back" to itself thereby leading to divergence, to overcome this problem they decay the sent out update; as against this, LAP-SGD uses no buffer and does not track updates as such, averaging the model with each peer, similar to L-SGD, helps each of the peers to adjust their optimization dynamics, (c) it is not known if the token based model averaging of HOGWILD!++ is sufficient for training DNNs where generalization is the core point of concern, in place of that we observed that our asynchronous averaging provides an effective protocol of synchronization and often results in improving the generalization, (d) comparing the HOGWILD!++ thread-clusters to concurrent processes on GPUs in LAP-SGD, the latter uses a dedicated process that performs averaging without disturbing the local gradient updates thereby maximally reducing the communication overhead, (e) finally, the convergence theory of LAP-SGD guarantees its efficacy for DNN training, which we demonstrated experimentally, by contrast, HOGWILD!++ does not have any convergence guarantee.

Recently, Wang et al. (2020) proposed Overlap-local-SGD, wherein they suggested to keep a model copy at each worker, very similar to HOGWILD!++, which is simultaneously averaged when sequential computation for multiple iterations happen locally. They showed by limited experiments that it reduced the communication overhead in a non-iid training case based on CIFAR-10, however, not much is known about its performance in general cases. The asynchronous partitioned gradient update of LPP-SGD derives from Kungurtsev et al. (2019), however, unlike them we do not use locks and our implementation setting is decentralized, thus scalable.

## 5 CONCLUSION

Picking from where Golmant et al. (2018) concluded referring to their findings: *"These results suggest that we should not assume that increasing the batch size for larger datasets will keep training times manageable for all problems. Even though it is a natural form of data parallelism for large-scale optimization, alternative forms of parallelism should be explored to utilize all of our data more efficiently"*, our work introduces a fresh approach in this direction to addressing the challenge.

In our experiments, we observed that the natural system-generated noise in some cases effectively improved the generalization accuracy, which we could not obtain using the existing methods irrespective of any choice of seed for random sampling. The empirical findings suggest that the proposed variant of distributed SGD has a perfectly appropriate place to fit in the horizon of efficient optimization methods for training deep neural networks. As a general guideline for the applicability of our approach, we would suggest the following: monitor the resource consumption of a GPU that trains a CNN, if there is any sign that the consumption was less than 100%, try out LAP-SGD and LPP-SGD instead of arduously, and at times unsuccessfully, tuning the hyperparameters in order to harness the data-parallelism. The asynchronous averaging protocol makes LAP-SGD and LPP-SGD specially attractive to settings with large number of workers.

There are a plethora of small scale model and dataset combinations, where the *critical batch size*– after which the returns in terms of convergence per wall-clock time diminish–is small relative to existing system capabilities (Golmant et al. (2018)). To such cases LAP-SGD and LPP-SGD become readily useful. Yet, exploring the efficiency of LAP-SGD and LPP-SGD to train at massive scales, where hundreds of GPUs enable training IMAGENET in minutes (Ying et al. (2018)), is an ideal future goal. We also plan to extend the proposed methods to combine with communication optimization approaches such as QSGD (Alistarh et al. (2017)).

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

# A  APPENDIX A: CONVERGENCE THEORY

## A.1  PROBABILISTIC ASSUMPTIONS GOVERNING THE ASYNCHRONOUS COMPUTATION

Now let us discuss the formalities of the asynchronous computation. We consider that the presence of local HogWild-like asynchronous computation introduces *stochastic* delays, i.e., at each stochastic gradient computation, the set of parameters at which the stochastic gradient is evaluated is random, it follows some distribution. Thus, considering that, in the general case,

$$v_{t,i}^{q,j} \in \cup_{k \in \{0,...,j\}} \{x_{s,i}^{q,k}\}_{s \in \{0,...,t\}}$$

we can now define a probability that this parameter view block is equal to each of these potential historical parameter values. To this end we define $\mathbb{I}_{s,k}^{t,i,q,j}$ as the event that block $i$ in the view for GPU $q$ at major iteration $j$ and minor iteration $t$ is equal to the actual parameter $x_{s,i}^{q,k}$, and $p_{s,k}^{t,i,q,j}$ is its probability. Now, $v_{t,i}^{q,j}$ could be, e.g., $x_{s,i}^{q,j-1}$ for some $s \in \{0,...,K^{j-1}\}$, i.e., it could have been evaluated at a parameter before the last averaging took place.

In the nonconvex setting, it would be entirely hopeless to bound $\|x_{s,i}^{q,j-1} - v_{t,i}^{q,j}\|$ in general, in this case we can simply hope that the objective function decrease achieved by iterations with a gradient computed after an averaging step outweighs the worst-case increase that takes place before it. In order to perform such an analysis, we will need bound the probability that this worst case occurs. In order to facilitate the notation for this scenario, let us define $x_{-1,i}^{q,j} = x_{K^{j-1}-1,i}^{q,j-1}$, $x_{-2,i}^{q,j} = x_{K^{j-1}-2,i}^{q,j-1}$, etc., and then $p_{-1,j}^{t,i,q,j}$ correspondingly. We can extend this analogously to delays from before two averaging steps, etc. Note that with this notation, $p_{l,j}^{t,i,q,j}$ is well defined for any $l \le t$, $l \in \mathbb{Z}$, of course as long as $|l| \le \sum_{k=0}^{j-1} K^k$.

In order to derive an expected decrease in the objective, we need to bound the probability of an increase, which means bounding the probability that the view is older than the previous averaging, which can be bounded by a bound on the probability that a particular read is more than some number $\tau$ iterations old. We thus make the following assumptions,

**Assumption A.1.** *It holds that,*

1. $p_{l,j}^{t,i,q,j} = 0$ *for* $l \le t - D$ *(Maximum Delay)*
2. *There exists* $\{p_\tau\}_{\tau \in \{1,...,D\}}$ *such that for all* $(q, j, t)$, *it holds that* $\mathbb{P}\left[\cup_i \mathbb{I}_{l,j}^{t,i,q,j}\right] \le p_{t-\tau}$ *for* $l = t - \tau$ *(Uniform Bound for Components' Delays)*

With these, we can make a statement on the error in the view. In particular it holds that,

$$\mathbb{E}\left[\left\|v_t^{q,j} - x_t^{q,j}\right\| \mid \cap_i \cup_{l \ge 0} \mathbb{I}_{l,j}^{t,i,q,j}\right] \le \alpha_{j,t,q}^2 B \tag{6}$$

for some $B > 0$. This bound comes from Section A.B.6 in Nadiradze et al. (2020). Thus, if the view is taken such that all components were read after the last averaging step then we can bound the error as given.

## A.2  PROOF OF MAIN CONVERGENCE THEOREM

Recall the major iterations $\bar{x}_j$ is defined as the value of the parameters after an averaging step has taken place, which is of course well-defined as every GPU will have the same set of values for the parameters. The update to $\bar{x}_j$ can be written as,

$$\bar{x}_{j+1} = \frac{1}{Q} \sum_{q=1}^{Q} \left[ \bar{x}_j - \sum_{t=0}^{K_j - 1} \alpha_{j,t,q} \tilde{\nabla}_{i(q,j,t)} f(v_t^{q,j}) \right] \tag{7}$$

Where we define, $\tilde{\nabla}_i f(v_t^{q,j})$ as the vector of size $n$ whose $i$'ith components are the calculated stochastic gradient vector defined at $v_t^{q,j}$, with the rest of the components padded with zeros. We indicate the component chosen $i(q, j, t)$ depends on the GPU and minor and major iteration, allowing for flexibility in the choice of block update (including the entire vector).

We are now ready to prove the convergence Theorem. The structure of the proof will follow Zhou & Cong (2017), who derives the standard sublinear convergence rate for local SGD in a synchronous environment for nonconvex objectives.

We begin with the standard application of the Descent Lemma,

$$
\begin{aligned}
f(\bar{x}_{j+1}) - f(\bar{x}_j) \leq &-\langle \nabla f(\bar{x}_j), \tfrac{1}{Q} \sum_{q=1}^{Q} \sum_{t=0}^{K_j-1} \alpha_{j,t,q} \tilde{\nabla}_{i(j,t,q)} f(v_t^{q,j}) \rangle \\
&+ \tfrac{L}{2} \left\| \tfrac{1}{Q} \sum_{q=1}^{Q} \sum_{t=0}^{K_j-1} \alpha_j \tilde{\nabla}_{i(j,t,q)} f(v_t^{q,j}) \right\|^2
\end{aligned}
\tag{8}
$$

Now, since $\mathbb{E} \tilde{\nabla}_{i(j,t,q)} f(v_t^{q,j}) = \nabla_{i(j,t,q)} f(v_t^{q,j})$,

$$
\begin{aligned}
&-\mathbb{E}\langle \nabla f(\bar{x}_j), \tilde{\nabla}_{i(j,t,q)} f(v_t^{q,j}) \rangle \\
&\quad = -\tfrac{1}{2} \left( \|\nabla_{i(j,t,q)} f(\bar{x}_j)\|^2 + \mathbb{E}\|\nabla_{i(j,t,q)} f(v_t^{q,j})\|^2 - \mathbb{E}\|\nabla_{i(j,t,q)} f(v_t^{q,j}) - \nabla_{i(j,t,q)} f(\bar{x}_j)\|^2 \right)
\end{aligned}
\tag{9}
$$

We now split the last term by the two cases and use equation 6,

$$
\begin{aligned}
&\mathbb{E}\|\nabla_{i(j,t,q)} f(v_t^{q,j}) - \nabla_{i(j,t,q)} f(\bar{x}_j)\|^2 \\
&\quad = \mathbb{E}\left[ \|\nabla_{i(j,t,q)} f(v_t^{q,j}) - \nabla_{i(j,t,q)} f(\bar{x}_j)\|^2 \,|\, \cap_i \cup_{l \geq 0} \mathbb{I}_{l,j}^{t,i,q,j} \right] \mathbb{P}\left[ \cap_i \cup_{l \geq 0} \mathbb{I}_{l,j}^{t,i,q,j} \right] \\
&\qquad + \mathbb{E}\left[ \|\nabla_{i(j,t,q)} f(v_t^{q,j}) - \nabla_{i(j,t,q)} f(\bar{x}_j)\|^2 \,|\, \left(\cap_i \cup_{l \geq 0} \mathbb{I}_{l,j}^{t,i,q,j}\right)^c \right] \mathbb{P}\left[ \left(\cap_i \cup_{l \geq 0} \mathbb{I}_{l,j}^{t,i,q,j}\right)^c \right] \\
&\quad \leq \alpha_{j,t,q}^2 B \, \mathbb{P}\left[ \cap_i \cup_{l \geq 0} \mathbb{I}_{l,j}^{t,i,q,j} \right] \\
&\qquad + 2\left( \|\nabla_{i(j,t,q)} f(\bar{x}_j)\|^2 + \mathbb{E}\|\nabla_{i(j,t,q)} f(v_t^{q,j})\|^2 \right) \mathbb{P}\left[ \left(\cap_i \cup_{l \geq 0} \mathbb{I}_{l,j}^{t,i,q,j}\right)^c \right]
\end{aligned}
$$

and thus combining with equation 22 we get the overall bound,

$$
\begin{aligned}
&-\mathbb{E}\langle \nabla f(\bar{x}_j), \tilde{\nabla}_{i(j,t,q)} f(v_t^{q,j}) \rangle \\
&\quad \leq -\tfrac{1}{2}\left( \|\nabla_{i(j,t,q)} f(\bar{x}_j)\|^2 + \mathbb{E}\|\nabla_{i(j,t,q)} f(v_t^{q,j})\|^2 - \alpha_{j,t,q}^2 B \right) \mathbb{P}\left[ \cap_i \cup_{l \geq 0} \mathbb{I}_{l,j}^{t,i,q,j} \right] \\
&\qquad + \tfrac{1}{2}\left( \|\nabla_{i(j,t,q)} f(\bar{x}_j)\|^2 + \mathbb{E}\|\nabla_{i(j,t,q)} f(v_t^{q,j})\|^2 \right) \mathbb{P}\left[ \left(\cap_i \cup_{l \geq 0} \mathbb{I}_{l,j}^{t,i,q,j}\right)^c \right] \\
&\quad \leq -\left( \|\nabla_{i(j,t,q)} f(\bar{x}_j)\|^2 + \mathbb{E}\|\nabla_{i(j,t,q)} f(v_t^{q,j})\|^2 \right) \mathbb{P}\left[ \cap_i \cup_{l \geq 0} \mathbb{I}_{l,j}^{t,i,q,j} \right] \\
&\qquad + \tfrac{1}{2}\left( \|\nabla_{i(j,t,q)} f(\bar{x}_j)\|^2 + \mathbb{E}\|\nabla_{i(j,t,q)} f(v_t^{k,j})\|^2 \right) + \tfrac{\alpha_{j,t,q}^2 B}{2}
\end{aligned}
\tag{10}
$$

It can be seen from this expression that we must have,

$$
\mathbb{P}\left[ \cap_i \cup_{l \geq 0} \mathbb{I}_{l,j}^{t,i,q,j} \right] \geq \frac{1}{2} + \delta
\tag{11}
$$

for some $\delta > 0$ to achieve descent in a expectation for iteration $j$ for a sufficient small stepsizes. Since we are taking the sum of such iterations, we must have, ultimately,

$$
\begin{aligned}
&\tfrac{1}{Q} \sum_{q=1}^{Q} \sum_{t=0}^{K_j-1} \alpha_{j,t,q} \left[ \mathbb{P}\left[ \cap_i \cup_{l \geq 0} \mathbb{I}_{l,j}^{t,i,q,j} \right] - \tfrac{1}{2} - \alpha_{j,t,q}\left( QK_j + \tfrac{\alpha_{j,t,q}B}{2} \right) \right] \mathbb{E}\|\nabla_{i(j,t,q)} f(v_t^{q,j})\|^2 \\
&\quad + \tfrac{1}{Q} \sum_{q=1}^{Q} \sum_{t=0}^{K_j-1} \alpha_{j,t,q} \left[ \mathbb{P}\left[ \cap_i \cup_{l \geq 0} \mathbb{I}_{l,j}^{t,i,q,j} \right] - \tfrac{1}{2} - \tfrac{\alpha_{j,t,q}^2 B}{2} \right] \mathbb{E}\|\nabla_{i(j,t,q)} f(\bar{x}_j)\|^2 \geq \hat{\delta}_j
\end{aligned}
\tag{12}
$$

with $\sum_{j=0}^{\infty} \hat{\delta}_j \geq f(\bar{x}_0) - f_m$, where recall that $f_m$ is a lower bound on $f$, in order to achieve asymptotic convergence. The standard sublinear SGD convergence rate is recovered with any choice with $\alpha_{j,t,q} = \Theta\left( \tfrac{1}{\sqrt{J}} \right)$ and thus ultimately $\hat{\delta}_j = \Omega\left( \tfrac{1}{\sqrt{J}} \right)$.

Let us now consider the quantity $\mathbb{P}\left[ \cap_i \cup_{l \geq 0} \mathbb{I}_{l,j}^{t,i,q,j} \right]$ in more detail and study how the nature of the concurrency affects the possibility and rate of convergence.

In particular notice that,

$$
\mathbb{P}\left[ \cap_i \mathbb{I}_{l,j}^{t,i,q,j} \right] \leq 1 - p_{t-\tau}
$$

for $l \geq t - \tau$. In general of course we expect this quantity to increase as $l$ is closer to $t$.

Consider two extreme cases: if there is always only one SG iteration for all processes for all major iterations $j$, i.e., $K_j \equiv 1$, *any* delay means reading a vector in memory before the last major iteration,

and thus the probability of delay greater than zero must be very small in order to offset the worst possible ascent.

On the other hand, if in general $K_j >> \tau$, then while the first $\tau$ minor could be as problematic at a level depending on the probability of the small delay times, for $t > \tau$ clearly the vector $v_t^{q,j}$ satisfies equation 6.

Thus we can sum up our conclusions in the following statements:

1. Overall, the higher the mean, variance, and thickness of the tails of the delay, the more problematic convergence would be,

2. The larger the quantity of local iterations each GPU performs in between averaging, the more likely a favorable convergence would occur.

The first is of course standard and obvious. The second presents the interesting finding that if you are running HogWild-type SG iterations on local shared memory, performing local SGD with a larger gap in time between averaging results in more robust performance for local SGD.

This suggests a certain fundamental *harmony* between asynchronous concurrency and local SGD, more "aggressive" locality, in the sense of doing more local updates between averaging, coincides with expected performance gains and robustness of more "aggressive" asynchrony and concurrency, in the sense of delays in the computations associated with local processes.

In addition, to contrast the convergence in regards to the block size, clearly the larger the block the faster the overall convergence, since the norms of the gradient vectors appear. An interesting possibility to consider is if a process can roughly estimate or predict when averaging could be triggered, robustness could be gained by attempting to do block updates right after an expected averaging step, and full parameter vector updates later on in the major iteration.

## A.3   CONVERGENCE - SIMPLER CASE

In reference to the classic local SGD theory in particular Stich (2018) for the strongly convex case and Zhou & Cong (2017) for the nonconvex case, we consider the simpler scenario wherein $i(q, j, t) = [n]$ and $v_{t,i}^{q,j} = x_{s,i}^{q,j}$ with $s \geq 0$ for all $v_{t,i}^{q,j}$, i.e., at no point are local updates computed at gradients evaluated at model components existing in memory prior to the last averaging step. We shall see the local asynchrony introduces a mild adjustment in the constants in the strongly convex case, relative to the classic result, and results in no change whatsoever in the nonconvex case.

### A.3.1   STRONGLY CONVEX CASE

The proof of convergence will be based on Stich (2018), the synchronous case.

The formalism of the argument changes to Algorithm 4. Note that this is functionally the same, and simply the concept of major iterations is dispensed with, except to define $K_j$.

---

1  Initialize $x_0^q$ for all $q$ **for** $t = 1, ..., T$ **do**
2      **for** *all* $q$ **do**
3          Let $x_t^q = x_{t-1}^q - \alpha_{t,q} \tilde{\nabla} f(v_t^q)$
4          **if** *(t MOD $\sum_{j=1}^{J} K_j = 0$) for some $J$* **then**
5              Let $\bar{x}_{t+1} = \frac{1}{Q} \sum_{q=1}^{Q} x_t^q$

---

The only change in the procedure is that the stochastic gradients are computed as evaluated at a vector $v_t^q$, so we shall see how the convergence results contrasts with Stich (2018) for synchronous averaging when computations are performed in this manner.

Let,

$$\begin{aligned}
\bar{x}_t &= \tfrac{1}{Q} \sum_{q=1}^{Q} x_t^q, \\
g_t &= \tfrac{1}{Q} \sum_{q=1}^{Q} \tilde{\nabla} f(x_t^q), \\
\bar{g}_t &= \tfrac{1}{Q} \sum_{q=1}^{Q} \nabla f(x_t^q), \\
\widehat{g}_t &= \tfrac{1}{Q} \sum_{q=1}^{Q} \tilde{\nabla} f(v_t^q), \\
\mathring{g}_t &= \tfrac{1}{Q} \sum_{q=1}^{Q} \nabla f(v_t^q)
\end{aligned}$$

We have, as in the proof of Lemma 3.1 Stich (2018)

$$\begin{aligned}
\|\bar{x}_{t+1} - x^*\|^2 &= \|\bar{x}_t - \alpha_t \widehat{g}_t - x^*\|^2 = \|\bar{x}_t - x^* - \alpha_t \mathring{g}_t\|^2 + \alpha_t^2 \|\widehat{g}_t - \mathring{g}_t\|^2 \\
&+ 2\alpha_t \langle \bar{x}_t - x^* - \alpha_t \mathring{g}_t, \mathring{g}_t - \widehat{g}_t \rangle
\end{aligned} \tag{13}$$

Continuing,

$$\begin{aligned}
\|\bar{x}_t - x^* - \alpha_t \mathring{g}_t\|^2 &= \|\bar{x}_t - x^*\|^2 + \alpha_t^2 \|\mathring{g}_t\|^2 - 2\alpha_t \langle \bar{x}_t - x^*, \mathring{g}_t \rangle \\
&= \|\bar{x}_t - x^*\|^2 + \tfrac{\alpha_t^2}{Q} \sum_{q=1}^{Q} \|\nabla f(x_t^q)\|^2 - \tfrac{2\alpha_t}{Q} \sum_{q=1}^{Q} \langle \bar{x}_t - v_t^q + v_t^q - x^*, \nabla f(v_t) \rangle \\
&= \|\bar{x}_t - x^*\|^2 + \tfrac{\alpha_t^2}{Q} \sum_{q=1}^{Q} \|\nabla f(x_t^q) - \nabla f(x^*)\|^2 \\
&- \tfrac{2\alpha_t}{Q} \sum_{q=1}^{Q} \langle v_t^q - x^*, \nabla f(v_t) \rangle - \tfrac{2\alpha_t}{Q} \sum_{q=1}^{Q} \langle \bar{x}_t - v_t^q, \nabla f(v_t) \rangle
\end{aligned}$$

Using Young's inequality and $L$-smoothness,

$$\begin{aligned}
-2\langle \bar{x}_t - v_t^q, \nabla f(v_t) \rangle &\leq 2L \|\bar{x}_t - v_t^q\|^2 + \tfrac{1}{2L} \|\nabla f(v_t^q)\|^2 \\
&= 2L \|\bar{x}_t - v_t^q\|^2 + \tfrac{1}{2L} \|\nabla f(v_t^q) - \nabla f(x^*)\|^2 \\
&\leq 2L \|\bar{x}_t - v_t^q\|^2 + (f(v_t^q) - f^*)
\end{aligned}$$

Applying this to the above estimate of $\|\bar{x}_t - x^* - \alpha_t \mathring{g}_t\|^2$, we get,

$$\begin{aligned}
\|\bar{x}_t - x^* - \alpha_t \mathring{g}_t\|^2 &\leq \|\bar{x}_t - x^*\|^2 + \tfrac{2\alpha_t L}{Q} \sum_{q=1}^{Q} \|\bar{x}_t - v_t^q\|^2 \\
&+ \tfrac{2\alpha_t}{Q} \sum_{q=1}^{Q} \left( \left( \alpha_t L - \tfrac{1}{2} \right) (f(v_t^q) - f^*) - \tfrac{\mu}{2} \|v_t^q - x^*\|^2 \right)
\end{aligned}$$

Let $\alpha_t \leq \tfrac{1}{4L}$ so $\left( \alpha_t L - \tfrac{1}{2} \right) \leq -\tfrac{1}{4}$. By the convexity of $\tfrac{1}{4}(f(x) - f(x^*)) + \tfrac{\mu}{2} \|x - x^*\|^2$,

$$\begin{aligned}
-\tfrac{2\alpha_t}{Q} \sum_{q=1}^{Q} &\left( \tfrac{1}{4}(f(v_t^q) - f^*) + \tfrac{\mu}{2} \|v_t^q - x^*\|^2 \right) \\
&\leq -\tfrac{2\alpha_t}{Q} \sum_{q=1}^{Q} \left( \tfrac{1}{4}(f(x_t^q) - f^*) + \tfrac{\mu}{2} \|x_t^q - x^*\|^2 \right) \\
&+ \tfrac{\alpha_t}{2Q} \sum_{q=1}^{Q} \left( \|v_t^q - x_t^q\| + 2\mu \|v_t^q - x_t^q\|^2 \right)
\end{aligned}$$

Putting this in equation 13 and taking expectations, we get,

$$\begin{aligned}
\mathbb{E}\|\bar{x}_{t+1} - x^*\|^2 &\leq (1 - \mu\alpha_t)\mathbb{E}\|\bar{x}_t - x^*\|^2 + \alpha_t^2 \mathbb{E}\|\mathring{g}_t - \widehat{g}_t\|^2 \\
&- \tfrac{\alpha_t}{2} \mathbb{E}(f(\bar{x}_t) - f^*) + \tfrac{2\alpha_t L}{Q} \sum_{q=1}^{Q} \|\bar{x}_t - v_t^q\|^2 \\
&+ \tfrac{\alpha_t}{2Q} \sum_{q=1}^{Q} \left( \|v_t^q - x_t^q\| + 2\mu \|v_t^q - x_t^q\|^2 \right)
\end{aligned} \tag{14}$$

By Assumption 2.1, we have,

$$\mathbb{E}\|\mathring{g}_t - \widehat{g}_t\|^2 = \mathbb{E}\|\frac{1}{Q} \sum_{q=1}^{Q} \left( \tilde{\nabla} f(v_t^q) - \nabla f(v_t^q) \right) \|^2 \leq \frac{\sigma^2}{Q} \tag{15}$$

We have that,

$$\begin{aligned}
\sum_{q=1}^{Q} \|v_t^q - x_t^q\| &\leq \sum_{q=1}^{Q} \|v_t^q - x_t^q\|_1 \\
&\leq \sum_{q=1}^{Q} \sum_{s=\max t-\tau, t_0}^{t-1} \alpha_s \|\tilde{\nabla} f(v_s^q)\|_1 \\
&\leq \alpha_t Q \tau \sqrt{n} G
\end{aligned} \tag{16}$$

and similarly,

$$\begin{aligned}
\sum_{q=1}^{Q} \|v_t^q - x_t^q\|^2 &\leq \sum_{q=1}^{Q} \sum_{s=\max t-\tau, t_0}^{t-1} \alpha_s^2 \|\tilde{\nabla} f(v_s^q)\|^2 \\
&\leq Q \alpha_t^2 \tau G^2
\end{aligned} \tag{17}$$

Letting index $t_0$ be such that $t - t_0 \leq H := \max\{K_j\}$ when averaging takes place, i.e. $\bar{x}_{t_0} = x_{t_0}^q$ for all $q$, we have,

$$
\begin{aligned}
&\frac{1}{Q} \sum_{q=1}^{Q} \mathbb{E}\|\bar{x}_t - v_t^q\|^2 \\
&= \frac{1}{Q} \sum_{q=1}^{Q} \mathbb{E}\|v_t^q - x_t^q + x_t^q - x_{t_0} - (\bar{x}_t - x_{t_0})\|^2 \\
&\leq \frac{2}{Q} \sum_{q=1}^{Q} \mathbb{E}\left[\|v_t^q - x_t^q\|^2 + \|x_t^q - x_{t_0} - (\bar{x}_t - x_{t_0})\|^2\right] \\
&\leq \frac{2}{Q} \sum_{q=1}^{Q} \mathbb{E}\|x_t^q - x_{t_0}\|^2 + 2\alpha_t^2 \tau G^2 \\
&\leq \frac{2}{Q} \sum_{q=1}^{Q} H\alpha_{t_0}^2 \sum_{s=t_0}^{t-1} \mathbb{E}\|\tilde{\nabla} f(x_s^q)\|^2 + 2\alpha_t^2 \tau G^2 \\
&\leq \frac{2}{Q} \sum_{q=1}^{Q} H^2 \alpha_{t_0}^2 G^2 \leq H^2 \alpha_{t_0}^2 G^2 + 2\alpha_t^2 \tau G^2 \\
&\leq 8H^2 \alpha_t^2 G^2 + 2\alpha_t^2 \tau G^2
\end{aligned}
\tag{18}
$$

where we use $\mathbb{E}\|X - \mathbb{E}X\|^2 = \mathbb{E}\|X\|^2 - \|\mathbb{E}X\|^2$ and equation 17 to go from the third to the fourth line.

Finally, putting equation 15, equation 18, equation 16 and equation 17 into equation 14 we get that,

$$
\begin{aligned}
\mathbb{E}\|\bar{x}_{t+1} - x^*\|^2 &\leq (1 - \mu\alpha_t)\mathbb{E}\|\bar{x}_t - x^*\|^2 + \alpha_t^2 \frac{\sigma^2}{Q} \\
&\quad -\frac{\alpha_t}{2}\mathbb{E}(f(\bar{x}_t) - f^*) + 16\alpha_t^3 L H^2 G^2 \\
&\quad +\frac{\alpha_t^2 \tau \sqrt{n} G}{2} + 2(\mu + 2L)\tau \alpha_t^3 G^2
\end{aligned}
$$

Finally, using Lemma 3.4 Stich (2018), we obtain, with $a > \max\{16\kappa, H\}$ for $\kappa = L/\mu$, and $w_t = (a + t)^2$,

$$
\begin{aligned}
&\mathbb{E}f\left(\frac{1}{QS_Q}\sum_{q=1}^{Q}\sum_{t=0}^{T-1} w_t x_t^q\right) - f^* \\
&\leq \frac{\mu a^3}{2S_T}\|x_0 - x^*\|^2 + \frac{4T(T+2a)}{\mu S_T}\left(\frac{\sigma^2}{Q} + \frac{\tau\sqrt{n}G}{2}\right) \\
&\quad + \frac{256T}{\mu^2 S_T}\left(16LH^2G^2 + 2(\mu + 2L)\tau G^2\right)
\end{aligned}
\tag{19}
$$

which simplifies to, using $\mathbb{E}\mu\|x_0 - x^*\| \leq 2G$,

$$
\begin{aligned}
&\mathbb{E}f\left(\frac{1}{QS_Q}\sum_{q=1}^{Q}\sum_{t=0}^{T-1} w_t x_t^q\right) - f^* \\
&= O\left(\frac{1}{\mu QT} + \frac{\kappa + H}{\mu QT^2}\right)\sigma^2 + O\left(\frac{\tau\sqrt{n}}{\mu T}\right)G \\
&\quad + O\left(\frac{\tau\sqrt{n}(\kappa + H)}{\mu T^2}\right)G + O\left(\frac{\kappa H^2 + \tau(\mu + 2L)}{\mu T^2} + \frac{\kappa^3 + H^3}{\mu T^3}\right)G^2
\end{aligned}
\tag{20}
$$

### A.3.2 NONCONVEX CASE

This proof will again follow Zhou & Cong (2017). In this case the step-sizes $\{\alpha_{(j}, t, q)\}$ are independent of $t$ and $q$, i.e., they are simple $\{\alpha_j\}$. Thus,

$$
\bar{x}_{j+1} = \frac{1}{Q}\sum_{q=1}^{Q}\left[\bar{x}_j - \sum_{t=0}^{K_j - 1} \alpha_j \tilde{\nabla} f(v_t^{q,j})\right]
$$

And thus,

$$
\begin{aligned}
f(\bar{x}_{j+1}) - f(\bar{x}_j) &\leq -\langle\nabla f(\bar{x}_j), \frac{1}{Q}\sum_{q=1}^{Q}\sum_{t=0}^{K_j - 1} \alpha_j \tilde{\nabla} f(v_t^{q,j})\rangle \\
&\quad + \frac{L}{2}\left\|\frac{1}{Q}\sum_{q=1}^{Q}\sum_{t=0}^{K_j - 1} \alpha_j \tilde{\nabla} f(v_t^{q,j})\right\|^2
\end{aligned}
\tag{21}
$$

Now, since $\mathbb{E}\tilde{\nabla} f(v_t^{q,\alpha}) = \nabla f(v_t^{q,\alpha})$,

$$
\begin{aligned}
&-\mathbb{E}\langle\nabla f(\bar{x}_j), \tilde{\nabla} f(v_t^{q,j})\rangle \\
&= -\frac{1}{2}\left(\|\nabla f(\bar{x}_j)\|^2 + \mathbb{E}\|\nabla f(v_t^{q,j})\|^2 - \mathbb{E}\|\nabla f(v_t^{q,j}) - \nabla f(x_j)\|^2\right) \\
&\leq -\frac{1}{2}\left(\|\nabla f(\bar{x}_j)\|^2 + \mathbb{E}\|\nabla f(v_t^{q,j})\|^2 - L^2\mathbb{E}\|v_t^{q,j} - \bar{x}_j\|^2\right)
\end{aligned}
\tag{22}
$$

We now continue the proof along the same lines as in Zhou & Cong (2017). In particular, we get

$$
\mathbb{E}\|v_t^{q,j} - \tilde{x}_j\|^2 \leq t^2\alpha_j^2\sigma^2 + t\alpha_j^2\mathbb{E}\sum_{s=0}^{t-1}\|\nabla f(v_s^{q,j})\|^2
$$

Let us define $\bar{K} = \max_j\{K_j\}$ and $\underline{K} = \min_j\{K_j\}$.

We now have,

$$-\alpha_j \sum_{t=0}^{K_j-1} \mathbb{E}\langle \nabla f(\bar{x}_j), \tilde{\nabla} f(v_t^{q,j})\rangle \leq -\frac{(\underline{K}+1)\alpha_j}{2}\left(1 - \frac{L^2\alpha_j^2 K(\bar{K}-1)}{2(\underline{K}+1)}\right)\|\nabla f(\tilde{x}_j)\|^2$$
$$-\frac{\alpha_j}{2}\left(1 - \frac{L^2\alpha_j^2(\bar{K}+1)(\bar{K}-2)}{2}\right)\sum_{t=0}^{K-1}\mathbb{E}\|\nabla f(v_t^{q,j})\|^2 + \frac{L^2\alpha_j^3\sigma^2(2\bar{K}-1)K(\bar{K}-1)}{12}$$

Similarly, it also holds that,

$$\frac{L}{2}\left\|\frac{1}{Q}\sum_{t=0}^{K_j-1}\alpha_j\tilde{\nabla}f(v_t^{q,j})\right\|^2 \leq \frac{LK_j^2\alpha_j^2\sigma^2}{2Q} + \frac{LK_j\alpha_j^2}{2}\sum_{t=1}^{K_j-1}\mathbb{E}\|\nabla f(v_t^{q,j})\|^2$$

And so, finally,

$$\mathbb{E}f(\bar{x}_{j+1}) - f(\bar{x}_j) \leq -\frac{(\underline{K}+1)\alpha_j}{2}\left(1 - \frac{L^2\alpha_j^2\bar{K}(\bar{K}-1)}{2(\underline{K}+1)} - \frac{L\alpha_j\bar{K}}{\underline{K}+1}\right)\|\nabla f(\tilde{x}_j)\|^2$$
$$-\frac{\alpha_j}{2}\left(1 - \frac{L^2\alpha_j^2(\bar{K}+1)(\bar{K}-2)}{2} - L\alpha_j\bar{K}\right)\sum_{q=1}^{Q}\sum_{t=1}^{K_j-1}\mathbb{E}\|\nabla f(v_t^{q,j})\|^2$$
$$+\frac{L^2\alpha_j^3\sigma^2(2\bar{K}-1)\bar{K}(\bar{K}-1)}{12} + \frac{LK^2\alpha_j^2\sigma^2}{2Q}$$

Now, if/once $\alpha_j$ is small enough such that,

$$1 \geq \frac{L^2\alpha_j^2(\bar{K}+1)(\bar{K}-2)}{2} + L\alpha_j\bar{K}$$

then the second term above disappears, and the result is exactly the same as in Zhou & Cong (2017). Specifically, if $1 - \delta \geq L^2\alpha_j^2$

$$\mathbb{E}\sum_{j=1}^{J}\frac{\alpha_j\|\nabla f(\bar{x}_j)\|^2}{\sum_{l=1}^{J}\alpha_j} \leq \frac{2(f(\tilde{x}_1)-F^*)}{(\underline{K}-1+\delta)\sum_{j=1}^{J}\alpha_j}$$
$$+\sum_{j=1}^{J}\frac{L\bar{K}\alpha_j^2 M}{\sum_{l=1}^{J}\alpha_l(\underline{K}-1+\delta)}\left(\frac{\bar{K}}{Q} + \frac{L(2\bar{K}-1)(\bar{K}-1)\alpha_j}{6}\right)$$

# B   APPENDIX B: AN ARGUMENT FOR INCREASED GENERALIZATION ACCURACY FOR LOCAL SGD

## B.1   WIDE AND NARROW WELLS

In general it has been observed that whether a local minimizer is shallow or deep, or how "flat" it is, seems to affect its generalization properties Keskar et al. (2019). Motivated by investigating the impact of batch size on generalization Dai & Zhu (2018) analyzed the generalization properties of SGD by considering the escape time from a "well", i.e., a local minimizer in the objective landscape, for a constant stepsize variant of SGD by modeling it as an overdamped Langevin-type diffusion process,

$$dX_t = -\nabla f(X_t)dt + \sqrt{2\epsilon}dW_t$$

In general "flatter" minima have longer escape times than shallow ones, where the escape time is the expectation in the number of iterations (defined as a continuous parameter in this sense) until the iterates leave the well to explore the rest of the objective landscape. Any procedure that increases the escape time for flatter minima as compared to shallower ones should, in theory, result in better generalization properties, as it is more likely then that the procedure will return an iterate that is in a shallow minimizer upon termination.

Denote with indexes $w$ for a "wide" valley local minimizer and $n$ for a "narrow" value, which also corresponds to smaller and larger minimal Hessian eigenvalues, respectively.

The work Berglund (2011) discusses the ultimately classical result that as $\epsilon \to 0$, the escape time from a local minimizer valley satisfies,

$$\mathbb{E}[\tau_e] = He^{C/\epsilon}$$

and letting the constant $H$ depend on the type of minimizer, it holds that that $H_w > H_n$, i.e., this time is longer for a wider valley.

We also have from the same reference,

$$\mathbb{P}[\tau_e > s\mathbb{E}[\tau_e]] = e^{-s}$$

## B.2 Averaging

We now contrast two procedures and compare the difference in their escape times for shallow and wider local minimizers. One is the standard SGD, and in one we perform averaging every $\tau_a$ time. In each case there are $Q$ processors, in the first case running independent instances of SGD, and in the other averaging their iterates. We model averaging probabilistically as it resulting in a randomized initialization within the well, and thus the escape time is a sequence of independent trials of length $\tau_a$ with an initial point in the well, i.e., escaping at time $\tau_e$ means that there are $Q \left\lceil \frac{\tau_e}{\tau_a} \right\rceil$ trials wherein none of the $Q$ sequences escaped within $\tau_a$, and then one of them escaped in the next set of $Q$ trials.

For ease of calculation, let us assume that $\tau_a = \frac{1}{2}\mathbb{E}[\tau_e^w] = 2\mathbb{E}[\tau_e^n]$, where $\tau_e^w$ and $\tau_e^n$ are the calculated single process escape time from a wide and shallow well, respectively.

If any one of the local runs escapes, then there is nothing that can be said about the averaged point, so a lack of escape is indicated by the case for which all trajectories, while periodically averaged, stay within the local minimizer value.

Now consider that if no averaging takes place, we sum up the probabilities for the wide valley that they all escape after time $(i-1)\tau$ time and, given that they do so, not all of them escape after $i\tau_a$.

$$\mathbb{E}[\tau_e^w] \leq \sum_{i=1}^{\infty} P(\tau_e^w > (i-1)\tau_a)^Q (1 - P(\tau_e^w > \tau_a i | \tau_e^w > (i-1)\tau_a)^Q) i \tau_a$$
$$\leq \sum_{i=1}^{\infty} \left( e^{-\frac{Q(i-1)}{2}} \left( 1 - e^{-\frac{Q}{2}} \right) \tau_a i \right)$$

For the narrow well this is,

$$\mathbb{E}[\tau_e^n] \geq \sum_{i=1}^{\infty} P(\tau_e^n > (i-1)\tau_a)^Q (1 - P(\tau_e^n > \tau_a i | \tau_e^n > (i-1)\tau_a)^Q)(i-1)\tau_a$$
$$\geq \sum_{i=1}^{\infty} \left( e^{-2Q(i-1)} \left( 1 - e^{-2Q} \right) \tau_a (i-1) \right)$$

The difference in the expected escape times satisfies,

$$\mathbb{E}[\tau_e^w - \tau_e^n] \leq \sum_{i=1}^{\infty} \left[ \left[ \left( e^{-\frac{Q(i-1)}{2}} \left( 1 - e^{-\frac{Q}{2}} \right) \right) - \left( e^{-2Q(i-1)} \left( 1 - e^{-2Q} \right) \right) \right] (i-1) \right.$$
$$\left. + \left( e^{-\frac{Q(i-1)}{2}} \left( 1 - e^{-\frac{Q}{2}} \right) \right) \right] \tau_a$$

Recall that in the case of averaging, if escaping takes place between $(i-1)\tau_a$ and $i\tau_a$ there were no escapes with less that $\tau_a$ for $M$ processors multiplied by $i-1$ times trials, and at least one escape between $(i-1)\tau_a$ and $i\tau_a$, i.e., not all did not escape between these two times.

The expected first escape time for any trajectory among $Q$ from a wide valley, thus, is,

$$\mathbb{E}[\tau_e^{a,w}] \leq \sum_{i=1}^{\infty} \left( \mathbb{P}[\tau_e^w > \tau_a]^{(i-1)Q} (1 - \mathbb{P}[\tau_e^w > \tau_a]^{Mi}) \tau_a i \right)$$
$$\leq \sum_{i=1}^{\infty} e^{-\frac{(i-1)Q}{2}} (1 - e^{-\frac{iQ}{2}}) \tau_a i$$

And now with averaging, the escape time from a narrow valley satisfies,

$$\mathbb{E}[\tau_e^{a,n}] \geq \sum_{i=1}^{\infty} \left( \mathbb{P}[\tau_e^n > \tau_a]^{(i-1)Q} (1 - \mathbb{P}[\tau_e^n > \tau_a]^{Qi}) \tau_a (i-1) \right)$$
$$\geq \sum_{i=1}^{\infty} e^{-2(i-1)Q} (1 - e^{-2iQ}) \tau_a (i-1)$$

With now the difference in the expected escape times satisfies,

$$\mathbb{E}[\tau_e^{a,w} - \tau_e^{a,n}] \leq \sum_{i=1}^{\infty} \left[ \left[ e^{-\frac{(i-1)Q}{2}} (1 - e^{-\frac{iQ}{2}}) \right. \right.$$
$$\left. \left. - e^{-2(i-1)Q}(1 - e^{-2iQ}) \right] (i-1) + e^{-\frac{(i-1)Q}{2}} (1 - e^{-\frac{iQ}{2}}) \right] \tau_a$$

It is clear from the expressions then that the upper bound for the difference is larger in the case of averaging. This implies that averaging results in a greater difference between the escape times between wider and shallow local minimizers, suggesting that, on average if one were to stop a process of training and use the resulting iterate as the estimate for the parameters, this iterate would more likely come from a flatter local minimizer if it was generated with a periodic averaging procedure, relative to standard SGD. Thus it should be expected, at least by this argument that better generalization is more likely with periodic averaging.

Note that since they are both upper bounds, this isn't a formal proof that in all cases the escape times are more favorable for generalization in the case of averaging, but a guide as to the mathematical intuition as to how this could be the case.

