# OpenReview forum: "Local SGD Meets Asynchrony"
_ICLR.cc/2021/Conference — Reject_

### Official Review · AnonReviewer3 · 2020-10-27
**This paper  proposes a class of  asynchronous local SGD by combining the local SGD with shared-memory based asynchronous gradient updates.**

**Rating:** 5
**Confidence:** 4

**Review:**

This paper  proposes a class of  asynchronous local SGD by combining the local SGD with shared-memory based asynchronous gradient updates. Moreover, it proves that the proposed methods guarantee ergodic convergence for non-convex objectives, and achieves the classic sub-linear rate under standard assumptions. Some experimental results verify the efficiency of the proposed methods.  Unfortunately, I do not find Appendix of the paper, so I can not judge the contribution of this paper.  I hope the authors to upload the supplementary materials.
In addition, because there are too many notations in this paper, it is easy to be confused. I hope the authors will introduce these notations in a table for readers.

---

> ### Author Response · Authors · 2020-11-13
> **Supplementary material**
>
> Thank you for your time. We had posted an anonymous link to the appendix as an official comment to the reviewers, PCs, and ACs in our submission. In the revision, we have uploaded the appendix and have updated the text with sufficiently defining the symbols.

---

### Official Review · AnonReviewer2 · 2020-10-29
**Official Blind Review #2**

**Rating:** 4
**Confidence:** 3

**Review:**

In this paper, the authors argue that the mini-batch method and local SGD method suffers generalization performance degradation for large local mini-batch size. An asynchronous method is proposed to improve the generalization performance. A sublinear convergence rate is provided for the non-convex objective. As there are some missing definitions and little explanation of the proposed method, the reviewer finds the paper hard to read.

1. The comparison shown in table 1 is problematic. When the local batch size is increased from 128 to 1024, the learning rate also needs to be scaled accordingly. It is not surprising that we observe performance issue if we do not increase the learning rate. This makes the motivation of the proposed algorithm questionable.

2. It is not clear to the reviewer about the intuition of why local asynchrony can mitigate the generalization issue. In particular, compared to local SGD, the asynchrony introduces additional staleness, which can hinder the convergence.

3. What are the definitions of a^q and U^q? These two symbols are used in many places in the paper. But there are no definitions of them. And, what is s_j^q in Algorithm 1b?

4. What is the motivation of having the process alternate between a full and a partitioned lock-free asynchronous gradient updates to the model $x^q$? It seems to the reviewer that you are switching between SGD and block coordinate descent. This can reduce computational complexity but worsen the convergence at the same time.

5. Theorem 2.1 does not show any improvement by using more workers.

---

> ### Author Response · Authors · 2020-11-13
> **(Mis-)reading Table 1, explanation of generalization, etc.**
>
> Thank you for your time and comments.
>
> * Comparison in Table 1: Kindly refer to the caption of Table 1 as in the submission, we scale the LR precisely as you mentioned. In particular, in this case, for an aggregated BS=2048 (128x16), after the first 5 warmup epochs, LR is 1.6 which is dampened at 150 and 225 epochs by a factor 0.1, which is standard. For the baseline BS=128, there is no warmup and LR is taken 0.1 initially.
>
> * Generalization:
> 1. First, please see the introduction of the paper. Asynchrony allows for the concomitant use of all computing resources like parallel batch SGD, however without employing large minibatch size. Thus our algorithm has better generalization properties than large batch parallel SGD, while having the desiderata of computational efficiency. On the other hand, relative to local SGD, it exhibits better end-to-end time performance of training, since it no longer requires computing resources to be idle. Thus, intuitively, our algorithm is meant to have the efficiency of large batch parallel SGD and the generalization of local SGD.
> 2. Above and beyond this, if you see the discussion section on Generalization and the associated part of the supplement, we have added an argument as to how local SGD and asynchrony contributes to better generalization. Note that although staleness hurts convergence, it has never been observed that it hurts generalization, and in this case, we have demonstrated how it can be considered as contributing additional injected noise, which is known to improve generalization.
>
> * Symbols: Thanks for the remarks. The compact definitions of symbols were mainly due to holding back some texts because of limited space. We have now added extra texts and have carefully defined each of the symbols.
>
> * On intermittent partitioned updates: Exactly, we are switching between SGD and block coordinate descent in the method LPP-SGD (not in LAP-SGD). Consequently, as noted by the reviewer it would result in reduced computational complexity but worsen the theoretical convergence at the same time. Evidently, the motivation of having the process alternate between a full and a partitioned lock-free asynchronous gradient updates to the model is empirically presented in Tables 3 and 6: we observe that we gain in performance without any loss in optimization quality. However, without the alternating methodology, purely block coordinate descent does not always achieve the same quality of optimization results as observed in [1].
>
> * Theorem 2.1 and number of worker: The paper should be considered in the space of parallel batch SGD and local SGD. In the first case, extra parallelization makes the gradient more accurate by effectively using a large batch size, which reduces variance but does not exhibit speedup per se. In the case of local SGD, there is no speedup exhibited in the convergence theory for nonconvex objectives. However, by enabling concurrency to avoid idle processing, our method is able to achieve speedup benefits vis-a-vis local SGD. How this can be seen from the statement of the main convergence theorem is a subtle point, and we have added a Remark to this effect in the revised version.
>
> [1] Asynchronous Optimization Methods for Efficient Training of Deep Neural Networks with Guarantees, Kungurtsev et al. 2019.

---

### Official Review · AnonReviewer4 · 2020-10-29
**Paper does not pass the bar (technically strengths and novelty aspects) for acceptance at ICLR**

**Rating:** 4
**Confidence:** 4

**Review:**

The paper proposes a method to address the "local scalability" problem in decentralized SGD. The authors study existing data parallel approaches and building upon the limitations, they propose two approaches - a synchronous and asynchronous non-blocking methods which aim to utilize compute resources optimally and avoid using large batch sizes.

I have a few comments:
- In the introduction, the paper mentions problems of finite-sum minimization form but it is missing the regularizer terms. Is this intended? It is much more common to see finite-sum minimization optimization with l1/l2 regularizers to control model complexity. I am bit surprised if the authors did not consider this.

- Degradation in accuracy upon increasing batch sizes is a popular problem and there has been lot of important work in this work (e.g. LARS [1], LAMB [2] to mention just a few). I don't see what the advantages of the proposed method are in comparison to these approaches which approach the problem differently but end up ultimately solving the large-batch accuracy issues. Can the authors comment this? on whether they investigated this empirically?

- From what I see, the proposed method does not "intelligently sample" the data points based on some structure/theoretical observation/heuristics etc. Correct me if I mis-understood. If so, how does proposed approach compare with other methods which aim to construct mini-batches or sub-sets of data on local workers more intelligently? such as [3] or [4]. I feel these related works also can extend to distributed decentralized SGD well.

- The scale of datasets is not large enough for distributed method. I would like to see experiments on datasets with larger # of data points, classes than what is currently presented.

- Did the authors perform any experiments on dense vs sparse datasets to see how the methods perform? The non-uniform distribution of data on various workers in my opinion can affect the optimization a lot and it would be good to see the authors comment on this.

- Table 2 is not convincing to me.
   (a) What if we go larger than 256 batch size? How does the accuracy look like in that case?
   (b) The gains are only marginal and don't seem significant. Did the authors perform any statistical significance tests or multiple random trials? If so, what was the variance of this experiment?

- The paper obtains a convergence rate of 1/\sqrt{T} based on what I understand? I am curious if the authors could comment on what assumptions need to be changed or what fundamentally stops the method from achieving a linear rate?

- Organization of paper can be improved. Minor: equation numbers are missing. The Quality/Perf column in Table 1 is confusing - what to quantify "ok", "good", "poor"?

I feel the paper does not meet the technical bar for an ICLR paper due to incomplete related work, unconvincing experiments. Proposed approaches do not seem very novel to me.


[1] https://arxiv.org/pdf/1708.03888.pdf

[2] https://arxiv.org/pdf/1904.00962.pdf

[3] https://negative-dependence-in-ml-workshop.lids.mit.edu/wp-content/uploads/sites/29/2019/06/SMDL_ICML_2019_ND_Workshop.pdf

[4] https://arxiv.org/pdf/1704.06731.pdf

---

> ### Author Response · Authors · 2020-11-13
> **On linear convergence, problem formulation, etc.**
>
> We thank the reviewer for taking the time to read the paper and comments.
> * Formulation without regularizer: We respectfully disagree with the statement "It is much more common to see finite-sum minimization optimization with l1/l2 regularizers to control model complexity" when it pertains to training DNNs. Adding a regularization term is indeed a technique that is commonly used to control model complexity, but in the case of training DNNs, there are a number of other techniques at one's disposal for doing so (see [2]). In general, literature in regards to training DNNs more commonly does not include such terms, especially in recent years, as the alternative techniques have grown to be more popular in practice. Most importantly, however, our paper is on the topic of local SGD, and, since the latest well-accepted work advocating this approach [1] does not include any regularization term in the objective, as well as every other paper, that we know of, that study local SGD and their variations, we believe it is more appropriate to not include this additional term.
> * On LARS: This is a good point. Indeed we performed experiments with LARS before our submission, however, due to limited space did not include those experiments as they did not provide any significantly different results than what we already reported in the submission. We have now revised the submission and included a paragraph on it in the Experiments section and a note on the same in the introduction.
> * "Intelligent" data sampling: Our data sampling is iid which is fairly standard in the literature on this topic, for example [1] and references thereof.
> * The scale of Dataset: We have included experiments on Imagenet training whereof training set includes 1.3 M samples and is a standard large scale image classification task.
> * Dense/sparse dataset: Again, our selected dataset -- CIFAR10/100, Imagenet, and sampling is standard on this topic.
> * Table 2: As mentioned, our entire proposition is to keep the local batch size small and perform concurrent updates to exploit compute resources. On increasing the batch size the training and test accuracy results degrade. We respectfully disagree with the conclusion that the gains are marginal: please notice that for a similar quality of result as the baseline, LAP-SGD provides up to 1.45x speedup without any additional hardware investment. Similarly, LPP-SGD provides up to 1.67x speedup without any extra hardware without compromising on the quality of results. The variance in experimental results is naturally present across the methods: for MB-SGD and PL-SGD by selecting different seeds and for the proposed methods due to the system-generated noise. However, as they are quite marginal, for example, a standard deviation of 3 seconds for an average latency of 1200 seconds, and almost identical across the methods, we opted to not include them to save space. We will definitely consider including them in the final version of the paper.
> * Linear convergence: As we are interested in training deep neural networks, the problems we consider are nonconvex and, due to the need for minibatch sampling, stochastic. There are no linearly convergent algorithms for either nonconvex or stochastic objectives. Furthermore, we have remarked on the speedup due to parallel and concurrent updates in the revised version.
> * Organization: Thanks for the comment. Most of these were due to space limits, which we have now addressed in the revised version.
>
> [1] DON’T USE LARGE MINI-BATCHES, USE LOCAL SGD, Lin et al. ICLR 2020.
> [2] Deep Learning, Goodfellow, et al.

---

### Decision · Program_Chairs · 2021-01-07
**Final Decision**

**Decision:**

Reject

**Comment:**

I think this paper has more positives than the reviews might indicate. And I do not share all the reviewers' concerns about the content of the paper. I think that there are a few concerns, though, that still suggest this paper should not be accepted as it is, when taken in conjunction with the concerns brought up by the reviewers.

On the positive side:

 - Asynchronous methods often give significant improvements, and the throughput benefits can even be seen here in Table 6.
 - The experiments are detailed with a lot of results comparing against many alternatives, including for ImageNet.

On the negative side:

 - The biggest concern I have with this paper is the scale of the experiments. This is supposedly about "distributed" SGD, but the largest-scale experiment was run on only two workstations, and many experiments were run in the S1 and S2 settings which don't seem to be distributed at all (run on a single workstation). That is, there's a mismatch between the scale at which these experiments were run and the scale at which people want to run distributed deep learning.
 - The theory seems to be only an incremental change to the standard local-SGD theory. The paper says "At a naive first glance, studying the convergence properties of locally asynchronous SGD would be an incremental to existing analyses for local SGD" but then it does not satisfactorily explain _why_ the approach is _not_ incremental. Not enough is done in the paper to explain why the analysis is not just a trivial combination of the local SGD with the standard approach to make an algorithmic analysis asynchronous. (Or, if the theoretical result _is_ incremental, the paper should make less of a big deal out of it.)
 - The description of the algorithm in Section 1.2 is confusing. I think it would benefit from being more concrete.

The paper should also compare against the paper "Asynchronous Decentralized Parallel Stochastic Gradient Descent" (Lian et al, 2017). It is actually not clear to me whether the method proposed here is a subset (or superset) of the method described in that paper, but they seem _very_ similar.